# Canady Cold Helios Plasma Reduces Soft Tissue Sarcoma Viability by Inhibiting Proliferation, Disrupting Cell Cycle, and Inducing Apoptosis: A Preliminary Report

**DOI:** 10.3390/molecules27134168

**Published:** 2022-06-29

**Authors:** Lawan Ly, Xiaoqian Cheng, Saravana R. K. Murthy, Olivia Z. Jones, Taisen Zhuang, Steven Gitelis, Alan T. Blank, Aviram Nissan, Mohammad Adileh, Matthew Colman, Michael Keidar, Giacomo Basadonna, Jerome Canady

**Affiliations:** 1Jerome Canady Research Institute for Advanced Biological and Technological Sciences, Takoma Park, MD 20912, USA; llawan@jcri-abts.com (L.L.); drxcheng@jcri-abts.com (X.C.); drsmurthy@jcri-abts.com (S.R.K.M.); ozjones@jcri-abts.com (O.Z.J.); 2Plasma Medicine Life Sciences, Takoma Park, MD 20912, USA; drtzhuang@jcri-abts.com; 3Section of Orthopedic Oncology, Rush University Medical Center, Chicago, IL 60612, USA; steven_gitelis@rush.edu (S.G.); alan.blank@rushortho.com (A.T.B.); matthewcolman@gmail.com (M.C.); 4The Department of General and Oncological Surgery—Surgery C, Sheba Medical Center, Ramat Gan 52621, Israel; aviram.nissan@sheba.health.gov.il (A.N.); mohammad.adileh@sheba.health.gov.il (M.A.); 5Department of Mechanical and Aerospace Engineering, The George Washington University, Washington, DC 20052, USA; keidar@gwu.edu; 6School of Medicine, University of Massachusetts, Worcester, MA 01655, USA; giacomo.basadonna@umassmed.edu; 7Department of Surgery, Holy Cross Hospital, Silver Spring, MD 20910, USA

**Keywords:** soft tissue sarcoma, liposarcoma, fibrosarcoma, synovial sarcoma, rhabdomyosarcoma, cold atmospheric plasma, cold plasma device, cancer treatment

## Abstract

Soft tissue sarcomas (STS) are a rare and highly heterogeneous group of solid tumors, originating from various types of connective tissue. Complete removal of STS by surgery is challenging due to the anatomical location of the tumor, which results in tumor recurrence. Additionally, current polychemotherapeutic regimens are highly toxic with no rational survival benefit. Cold atmospheric plasma (CAP) is a novel technology that has demonstrated immense cancer therapeutic potential. Canady Cold Helios Plasma (CHCP) is a device that sprays CAP along the surgical margins to eradicate residual cancer cells after tumor resection. This preliminary study was conducted in vitro prior to in vivo testing in a humanitarian compassionate use case study and an FDA-approved phase 1 clinical trial (IDE G190165). In this study, the authors evaluate the efficacy of CHCP across multiple STS cell lines. CHCP treatment reduced the viability of four different STS cell lines (i.e., fibrosarcoma, synovial sarcoma, rhabdomyosarcoma, and liposarcoma) in a dose-dependent manner by inhibiting proliferation, disrupting cell cycle, and inducing apoptosis-like cell death.

## 1. Introduction

Sarcoma is one of the major types of cancer and refers to tumors derived from mesenchymal or neural crest cells [1]. Accounting for 20% and <1% of all pediatric and adult solid tumors, respectively, sarcomas are exceedingly rare [2,3]. Sarcomas are also extremely heterogeneous with over 50 different histological subtypes, many of which can occur at any age or anatomical location [3]. Soft tissue sarcomas (STS), one of the two major subclasses of sarcoma, arise from diverse connective tissue and are more frequently diagnosed than bone sarcomas, which arise from bone or cartilage [4,5]. Due to its rarity and heterogenicity, it is likely that STS incidences are underestimated and initially misdiagnosed which could adversely affect patient outcomes [6,7,8].

Many STS subtypes are highly malignant and aggressive, more so in adults than in children. For example, adult patients with rhabdomyosarcoma (derived from skeletal muscle cells), synovial sarcomas (with partial epithelial differentiation), and fibrosarcoma (originating from tendon and fascia), have substantially lower five-year overall survival rates and higher risk of recurrence or metastasis than pediatric patients [9,10,11,12,13,14,15,16]. This survival discrepancy between adults and children is often attributed to reduced radio- or chemosensitivity with increasing age [9,11,13,15].

For the past several decades, surgery has been the first-line treatment for STS [17]. To mitigate the risk of tumor recurrence, the goal is to achieve a negative margin status, a term used to define a tumor-free surgical margin. Depending on the anatomical location of the tumor, achievement of negative margin status can be limited by anatomical constraints at the cost of healthy tissue and function loss. Since STS tumors present as painless enlarging masses which can eventually impede limb, organ, and nerve functions, tumors of the head/neck and extremities are often detected earlier and at smaller sizes than tumors of the trunk [18]. Tumor growth in the retroperitoneum, referred to the area behind the abdominal cavity, can be especially problematic as undetected growth can result in large tumors which can be challenging to resect [18,19].

Due to the difficulty of obtaining wide surgical margins near vital organs, positive microscopic margins, defined by residual disease within <1 mm of the surgical margin, are often a consequence of retroperitoneal STS [20]. Retroperitoneal STS tumors have the highest rate of positive microscopic margins (45%) compared to STS tumors of the head or neck (30%) and extremities (19%) [21]. Liposarcoma (LS), one of the most common STS subtypes [5,22], often occurs in the adipose tissue of the retroperitoneum as a low-grade tumor, also known as well-differentiated LS [23]. However, multiple local recurrences after inadequate resections can cause transition into a high-grade tumor with aggressive metastatic potential, i.e., dedifferentiated LS [23]. Consequently, positive microscopic margins of retroperitoneal LS reduce patient survival probability by half [20].

Adjuvant radio- and chemotherapies can be effective modalities for ensuring control of residual disease but are rarely beneficial to STS. Radiotherapies often fail to improve local control, making reresection necessary [19,24]. Chemotherapeutic options, such as doxorubicin, ifosfamide, and trabectedin, serve as palliative treatments since overall response rates are poor [25,26,27,28,29,30,31,32]. Polychemotherapeutic approaches offer higher overall response rates yet increase toxicity without any significant benefit to overall survival [25,26,27,28]. Despite current STS treatment options, margin status remains a significant predictor of distance recurrence-free survival and disease-specific survival [20].

Cold atmospheric plasma (CAP) is a novel and a relatively new technology with various biomedical applications, including cancer therapeutics [33,34]. CAP is a partially ionized gas, such as helium, nitrogen, or argon, and is composed of reactive oxygen and nitrogen species (RONS), which have been largely acknowledged to induce apoptosis in cancer cells [35]. The Canady Cold Helios Plasma (CHCP) System was the first CAP device to complete an FDA-approved phase 1 clinical trial for solid tumors (IDE G190165). Upon tumor resection, surgical margins were sprayed with CHCP to eradicate residual cancer cells for the prevention of tumor recurrence. CHCP is an advantageous adjuvant treatment since it is well-tolerated (26–30 °C) and does not cause thermal or physical damage to normal tissue [36,37]. It is primarily understood that CHCP induces apoptosis in cancer cells through 8-oxoG modification of histone RNA, degradation of histone RNA, and chromatin destabilization during S-phase [38]. As demonstrated in our previous studies, CHCP treatment reduced cell viability by 80–99% across numerous carcinoma cell lines (e.g., renal adenocarcinoma, colorectal carcinoma, pancreatic adenocarcinoma, ovarian adenocarcinoma, esophageal adenocarcinoma, and multiple breast adenocarcinomas) [36,37,39].

The purpose of this study was to evaluate the efficacy of CHCP on sarcomas, specifically STS. We examined the effects of CHCP on the cell viability of four human STS cell lines, each representing a different STS subtype shown in Table 1. A series of subsequent experiments were performed on liposarcoma cells to determine how CHCP inhibits STS cell viability (e.g., proliferation, cell cycle, and apoptosis). This focused investigation on liposarcoma was conducted in preparation for a humanitarian compassionate use case on a recurrent retroperitoneal myxoid liposarcoma patient in 2019 at Sheba Medical Center, Tel HaShomer, Israel. The cell line 94T778, derived from a recurrent well-differentiated retroperitoneal liposarcoma tumor, was the closest representing cell line to the patient tumor available. This preliminary study was necessary to establish the appropriate CHCP dose for the compassionate use case and later served as a reference for the treatment of STS patients in the FDA-approved phase 1 clinical trial from 2020–2021 at Sheba Medical Center, Tel HaShomer, Israel, and Rush University Medical Center, Chicago, IL, USA. The reports on the compassionate use case and clinical trial are currently under preparation for publication and will expand on the results of this preliminary report.

## 2. Results

### 2.1. Cell Viability

To determine the appropriate eradication dose for each STS subtype, MTT assays were performed 48-h post-CHCP treatment on all four STS cell lines (Figure 1 and Figure 2). In Figure 1A, CHCP did not reduce cell viability in a clear dose-dependent manner, likely because a helium flow rate of 1 L per min (LPM) was too weak to yield consistent results. However, when a helium flow rate of 3 LPM was used, CHCP reduced the viability of HT-1080, SW982, and RD cells in a dose-dependent manner compared to mock controls, demonstrated in Figure 1B. The highest tested CHCP dose of 120 power (P) for 2 min significantly reduced HT-1080 and SW982 viability by 95% and RD viability by 98% (Figure 1B). In general, HT-1080 cells had a greater susceptibility to CHCP, followed by SW982 and RD cells in that order.

Similarly, Figure 2 showed that CHCP reduced the cell viability of 94T778 cells in a dose-dependent manner compared to mock controls. A treatment of 120 P for 3 min was the minimal dose required to significantly reduce liposarcoma cell viability by 13% whereas the maximum treatment of 120 P for 7 min yielded a 93% reduction in viability. Compared to Figure 1, which utilized CHCP doses of 1 or 3 LPM at 20–120 P for 1–2 min, higher CHCP doses of 3 LPM at 120 P for 1–7 min were administered to 94T778 cells since its seeding cell density was greater to accommodate for subsequent experiments, which required higher cell counts, and to prepare for in vivo applications. For all cell lines tested in this study, power and treatment duration were significant factors in the reduction of cancer cell viability (*f* ≤ 0.033 and ** *f* ≤ 0.002381) (Appendix A). A series of subsequent experiments were performed on 94T778 cells to determine how CHCP inhibits STS cell viability.

### 2.2. Cell Proliferation

Expression of Ki67 proliferative marker was investigated 6-, 24-, and 48-h post-CHCP treatment, shown in Figure 3. Compared to mock controls (Figure 3A), Ki67 expression initially spiked in 94T778 cells 6-h post-CHCP treatment (Figure 3B), suggesting a possible survival mechanism and warranting further studies. However, Ki67 expression was eventually exhausted after 24 and 48 h (Figure 3C,D), indicating inhibition of cellular proliferation.

### 2.3. Cell Cycle

Cell cycle was continuously monitored for 48 h following CHCP treatment, depicted in Figure 4. CHCP doses of 120 P for 5 min and 120 P for 7 min caused an increase in the number of 94T778 cells in S/G2/M phase, indicating cell cycle arrest. Evidently, treatment for 5 min was not enough to induce permanent arrest as cells began to recover and resume proliferation after 24–48 h. However, treatment for 7 min was adequate to initiate cell death after 24 h, eliminating all live cells by 36–48 h.

### 2.4. Apoptosis

To confirm the CHCP-induced apoptosis, cells were stained for FITC Annexin V and Propidium Iodide (PI), and live, early apoptotic, and late apoptotic/dead cell populations were quantified by flow cytometry 24- and 48-h post-CHCP treatment, shown in Figure 5. CHCP treatment conditions were selected based on the viability data in Figure 2; treatments of 120 P for 7 min (high dose), 120 P for 5 min (medium dose), and 80 P for 5 or 7 min (low doses- not demonstrated by MTT) were selected to represent varying dose intensities of CHCP (Figure 5). Although a medium dose of 120 P for 5 min had no significant effect on the distribution of cell populations after 24 h, the live cell population significantly decreased to 64% by 48 h (Figure 5B). A high dose of 120 P for 7 min significantly reduced the live cell population from 93% to 7% after only 24 h (Figure 5B). And by 48 h, <1% of cells were alive, 12% were early apoptotic and 88% were late apoptotic/dead (Figure 5B). Altogether, this demonstrated that apoptotic cell death is dependent on CHCP dose intensity and incubation time after treatment administration.

## 3. Discussion

To the authors knowledge, this is the first study to compare the efficacy of CAP across multiple STS cell lines. Fibrosarcoma had the greatest susceptibility to CHCP, followed by synovial sarcoma, and rhabdomyosarcoma. Differences in CHCP sensitivity were also observed in our previous study which compared the efficacy of CHCP across multiple carcinoma cell lines; renal adenocarcinoma was highly susceptible to CHCP treatment whereas ovarian adenocarcinoma was relatively more resistant and required a higher dose for eradication [39]. Further studies are required to determine how molecular differences between cell lines or cancer subtypes contribute to CHCP sensitivity.

Initially identified in our previous studies on carcinoma cell lines [38,43], inhibition of Ki67 expression and initiation of apoptosis by CHCP was replicated in this study on liposarcoma cells. We also observed a spike in the number of liposarcoma cells in the S/G2/M phase followed by initiation of cell death, consistent with our recent study on breast cancer cell lines [38]. This previous study demonstrated that CHCP permanently disrupts the cell cycle through specific 8-oxoG modification of histone mRNA during the early S-phase. This compromises the stability of histone mRNA, leading to chromatin destabilization and apoptotic cell death [38]. Although this was demonstrated in breast cancer cells, it is possible that CHCP also induces oxidation of histone mRNA in liposarcoma cells, which will require confirmation.

In our previous studies for the establishment of CHCP dosimetry, cancer cells were seeded in concentrations of 1.0 × 10^5^ cells/well 24 h prior to CHCP treatment [39]. However, for this study, authors were required to lower the seeding concentration of 94T778 cells to 2.5 × 10^4^ cells/well because CHCP had no effect on 94T778 cell viability at higher seeding densities. This could be because of its unusually large cell size and its proportionately higher amount of histone mRNA, which would require a higher dose of CHCP to degrade. It was also observed that 94T778 grew relatively slower than other cancer cell lines, suggesting that slow-growing cells would be less susceptible to CHCP since histone mRNA would less likely be exposed during S-phase. However, other morphological or molecular differences could also play a role in 94T778 resistance. Determining these would require a stand-alone study of its own.

Moreover, in vitro follow-up studies will consider differential gene regulation as a mechanism of survival since we recently discovered BCL2A1 to be a key gene for CHCP resistance in breast cancers [44]. We anticipate that screening a panel of genes, associated with apoptosis (e.g., BCL2A1, TNF-α) and oxidative stress (e.g., APOE), will reveal therapeutic targets to increase STS susceptibility to CHCP for optimization of STS management.

Finally, based on our preliminary report, CHCP treatment of 120P for 7 min was determined to be the eradication dose for liposarcoma and would be the standard for future animal and clinical studies for STS.

## 4. Materials and Methods

All experiments were performed at the Jerome Canady Research Institute for Advanced Biological and Technological Sciences (JCRI-ABTS), Takoma Park, MD, USA.

**Cold Plasma Device.** CHCP was developed at JCRI-ABTS (U.S. Patent No. 9,999,462). The CHCP scalpel generates CAP through a USMI SS-601 MCa high-frequency electrosurgical generator and a USMI Canady Cold Plasma Conversion Unit. In-depth description and schematics of plasma generation by CHCP was detailed in our previous study [36]. To establish optimal treatment conditions for each cell line, a range of CHCP doses were tested: helium flow rate of 1 or 3 L per min (LPM); power settings of 20–120 P; treatment duration of 1–7 min. The power settings of 20 P, 40 P, 60 P, 80 P, 100 P, and 120 P yield powers deposited into the cold plasma discharge of 5 W, 8 W, 11 W, 15.7 W, 22.3 W, and 28.7 W at 3 lpm, respectively, and 5 W, 6 W, 7 W, 8 W, 9W and 11 W at 1 lpm, respectively [36]. Mock and helium only (0 P) controls were included in all experiments. CHCP treatment was performed in a laminar flow tissue culture hood at room temperature. The CHCP scalpel tip was fixed 1.5 cm above the cell culture media and remained unmoved for the duration of the treatment.

**Cell Culture.** Connective tissue fibrosarcoma (HT-1080), synovial sarcoma (SW-982), rhabdomyosarcoma (RD), and well-differentiated retroperitoneal liposarcoma (94T778) human cell lines were purchased from ATCC (Manassas, VA, USA). Following ATCC’s protocol, cells were cultured in either Eagle’s Minimum Essential Medium, Dulbecco’s Modified Eagle’s Medium, or Roswell Park Memorial Institute-1640 Medium supplemented with 10% fetal bovine serum (Sigma-Aldrich, St. Louis, MO, USA) and 1% Pen Strep (Thermo Fisher Scientific, Waltham, MA, USA) in a 37 °C and 5% CO_2_ humidified incubator (Thermo Fisher Scientific). When cells reached approximately 80% confluence, HT-1080, SW-982, and RD were seeded at a concentration of 5.0 × 10^3^ cells/well into 96-well plates (USA Scientific, Ocala, FL, USA) and were treated with 1 or 3 LPM at 20–120 P for 1–2 min for cell viability assays. Additionally, 94T778 was seeded at a concentration of 2.5 × 10^4^ cells/well into 12-well plates (USA Scientific) and were treated with 3 LPM at 120 P for 1–7 min for cell viability, proliferation, cell cycle, and apoptosis assays.

**Cell Viability Assay.** Thiazolyl blue tetrazolium bromide (MTT) assays were performed on all four cell lines 48-h post-CHCP treatment. Reagents were purchased from Sigma-Aldrich (St. Louis, MO, USA) and assays were performed following manufacturer’s protocol. The absorbance of the dissolved compound was measured by a BioTek Synergy HTX (Winooski, VT, USA) microplate reader at 570 nm.

**Cell Proliferation Analysis.** Images of immunofluorescent Ki67 expression in CHCP-treated 94T778 cells were obtained to detect changes in cell proliferation 6-, 24-, and 48-h post-CHCP treatment. Cells were seeded onto fibronectin-coated (Sigma-Aldrich) 12 mm diameter round cover glass (Thermo Fisher Scientific) in 12-well plates. Following CHCP treatment (3 LPM at 120 P for 5 min), cells were washed, fixed, and stained with Alexa Fluor 488 conjugated Ki-67 Rabbit mAb (Cell Signaling Technology, Danvers, MA, USA) and Isotype Control (Cell Signaling Technology) in a 1:200 dilution. After overnight incubation in 4 °C, the round cover slides with cells were transferred onto 1″ × 3″ × 1 mm microscope slides. Cells were covered with Antifade Mounting Reagent with DAPI (Vector Laboratories, Burlingame, CA, USA) before finally covered by a 24 × 50 mm cover glass (Cancer Diagnostics, Durham, NC, USA) and cured for 2 nights in 4 °C. Slides were imaged using Confocal LSM 510 (Carl Zeiss, Jena, Germany) with a 63 × lens and 405 and 488 nm laser bands.

**Cell Cycle Analysis.** The IncuCyte S3 Live-Cell Analysis System (Sartorius, Aubagne, France) continuously monitored cell cycle progression of 94T778 stable cell line every hour and up to 48 h following CHCP treatment (3 LPM at 120 P for 5 or 7 min). The stable cell line was generated using the IncuCyte^®^ Cell Cycle Green/Red Lentivirus Reagent (Sartorius).

**Apoptosis Assay.** Flow cytometry was used to detect apoptosis in 94T778 cells 24- and 48-h post-CHCP treatment. CHCP doses of 3 LPM at 80 P or 120 P for 5 or 7 min were selected. All live attached and dead detached cells were collected, washed, and stained with FITC Annexin V (BD Biosciences, Franklin Lakes, NJ, USA) and Propidium Iodide (Thermo Fisher Scientific) according to the manufacturer’s protocol. Early and late apoptosis was quantified and analyzed using FCS Express (De Novo Software, Pasadena, CA, USA).

**Statistics.** MTT assays were repeated 3 times with 2 replicates for each condition. Flow cytometry was repeated 3 times with 1 replicate for each condition. Data was plotted by Microsoft Excel 2016 as the mean ± standard error of the mean. A student *t*-test or a one-way analysis of variance (ANOVA) was used to check statistical significance where applicable. The differences were considered statistically significant for ^a^ *p* ≤ 0.05 or ^b^ *p* ≤ 0.05. A post-hoc test was used to check statistical significance where applicable. The differences were considered statistically significant for * *f* ≤ 0.0033 and ** *f* ≤ 0.002381. Confocal microscopy and IncuCyte experiments were each performed 1 time with 2 replicates for visualization, so statistics were not considered for these experiments.

## 5. Conclusions

Overall, our study strongly demonstrated CHCP to be a promising adjuvant for STS treatment. The in vitro preliminary data showed that CHCP effectively reduced STS viability by inhibiting proliferation, disrupting cell cycle progression, and inducing apoptosis. The next report will reveal the results of our gene expression studies on ex vivo STS tumor samples from our completed FDA-approved phase 1 clinical trial to elucidate the CHCP mechanism.

## Figures and Tables

**Figure 1 molecules-27-04168-f001:**
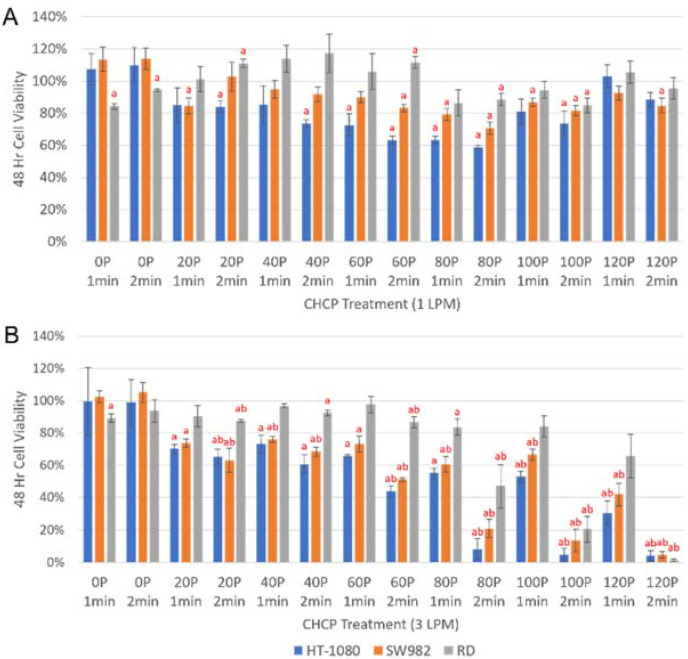
Bar graph showing the cell viability of HT-1080, SW982, and RD cells 48-h post-CHCP treatment compared to mock controls. Cells were seeded in 96-well plates at a concentration of 5 × 10^3^ cells/well and treated with (**A**) 1 or (**B**) 3 LPM at 20–120 P for 1–2 min. Helium alone (0 P) did not significantly impact HT-1080 or SW982 cell viability. CHCP significantly reduced HT-1080 and SW982 viability at all 3 LPM tested doses and RD viability at most 3 LPM tested doses compared to mock controls. Statistical significance for CHCP versus mock controls (^a^ *p* ≤ 0.05) and 1 LPM versus 3 LPM CHCP treatment (^b^ *p* ≤ 0.05) were considered.

**Figure 2 molecules-27-04168-f002:**
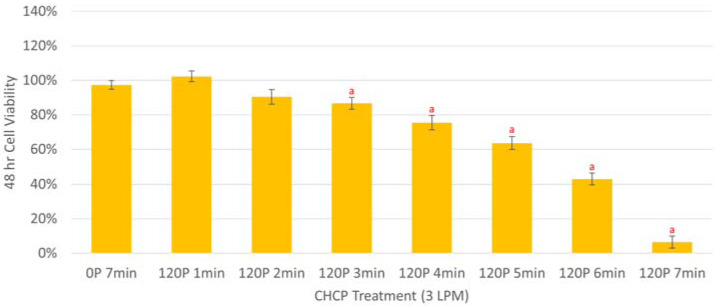
Bar graph showing the cell viability of 94T778 cells 48-h post-CHCP treatment compared to mock controls. Liposarcoma cells were seeded in 12-well plates at a concentration of 2.5 × 10^4^ cells/well and treated with 3 LPM at 120 P for 1–7 min. Helium alone (0 P) did not significantly impact cell viability. All treatment durations lasting at least 3 min significantly reduced viability of 94T778 cells compared to mock controls. Statistical significance for CHCP versus mock controls (^a^ *p* ≤ 0.05) were considered.

**Figure 3 molecules-27-04168-f003:**
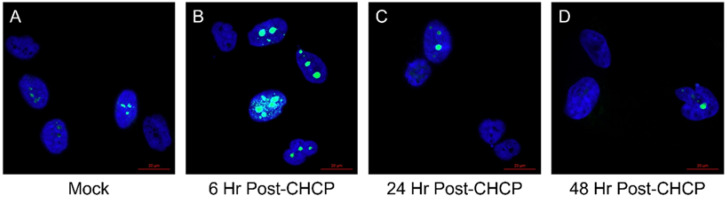
Representative confocal microscopy images showing Ki67 (green) expression in DAPI-stained (blue) (**A**) mock control and CHCP-treated (120 P for 5 min) liposarcoma cells after (**B**) 6, (**C**) 24, and (**D**) 48 h. Most mock control cells expressed Ki67, indicating normal cancer proliferation. CHCP-treated cells showed an initial spike in Ki67 expression after 6 h, followed by a sharp decrease in Ki67 expression after 24 and 48 h.

**Figure 4 molecules-27-04168-f004:**
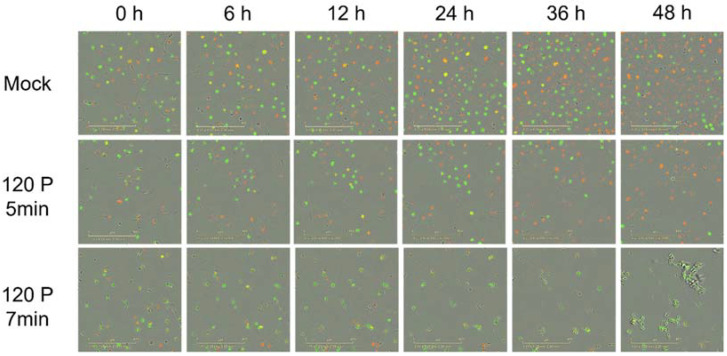
Representative IncuCyte images showing mock control and CHCP-treated cells in G1 (green), S/G2/M (red) and G2-S (yellow) phases over 48 h. Mock control had a relatively even population distribution of all cell cycle phases. Cells treated with 120 P for 5 min arrested in S/G2/M (6 h) before recovering (24–36 h). Cells treated with 120 P for 7 min immediately arrested in S/G2/M (0–6 h) before shrinking (12 h) and undergoing apoptosis (24 h), eventually resulting in zero visible live cells (36–48 h).

**Figure 5 molecules-27-04168-f005:**
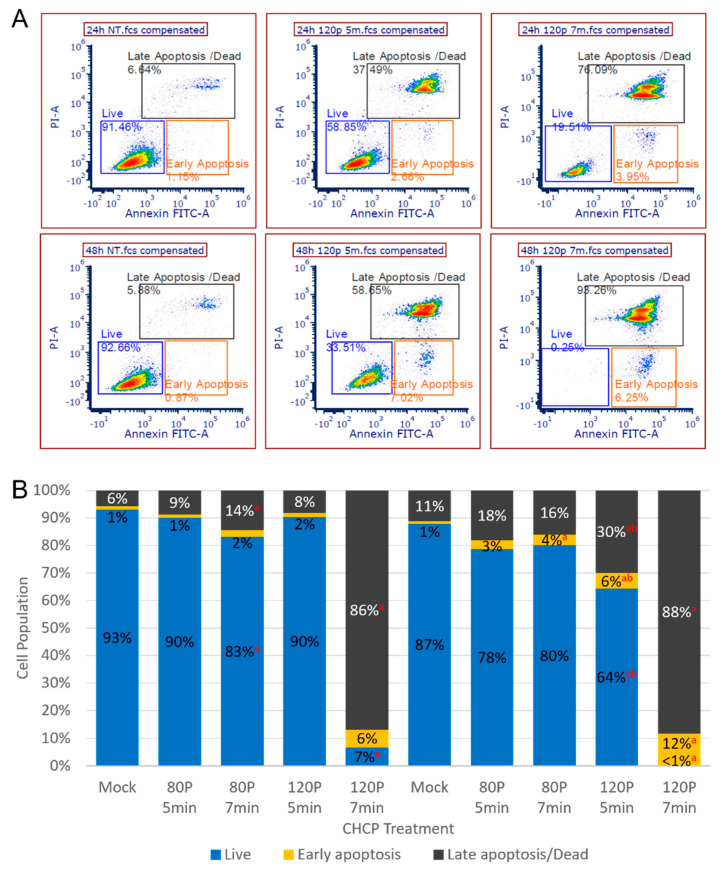
Distribution of cell population in live, early apoptotic, or late apoptotic/dead stages 24- or 48-h post-CHCP treatment presented in (**A**) flow cytometry images and (**B**) a bar graph. Statistical significance for CHCP versus mock controls (^a^ *p* ≤ 0.05) and 24- versus 48-h post-CHCP treatment (^b^ *p* ≤ 0.05) were considered.

**Table 1 molecules-27-04168-t001:** A list of the four human STS cell lines utilized in this study.

Human Cell Line	STS Subtypes	Tissue of Origin	References
HT-1080 ^1^	Fibrosarcoma	Connective	[14]
SW-982 ^1^	Synovial Sarcoma	Joint; Synovium	[40]
RD ^1^	Rhabdomyosarcoma	Muscle	[41]
94T778 ^1,2^	Liposarcoma	Adipose	[42]

^1^ Viability assays were performed in this study. ^2^ Proliferation, cell cycle, and apoptosis assays were performed in this study.

## Data Availability

All data and materials used in analysis are available upon reasonable request.

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
