# Peer review of "Canady Cold Helios Plasma Reduces Soft Tissue Sarcoma Viability by Inhibiting Proliferation, Disrupting Cell Cycle, and Inducing Apoptosis: A Preliminary Report"

_molecules, 2022, doi:10.3390/molecules27134168_

Round 1
Reviewer 1 Report
The paper deals with the study of the effects of plasma processing on soft tissue sarcoma cell lines. In the opinion of the reviewer, the paper is well organized and clearly explained except for some points that were missed. Thus, the reviewer suggests publishing the paper after major revisions as listed below:
- > In the Abstract the authors claim that “CHCP treatment reduced the viability of four different STS cell lines (i.e., fibrosarcoma, synovial sarcoma, rhabdomyosarcoma, and liposarcoma) in a dose-dependent manner by inhibiting proliferation, disrupting cell cycle, and inducing apoptosis-like cell death”. However, the proliferation, cell cycle, and apoptosis studies have been performed only on the cell line, 94T778 as clearly stated in Table 1. The reviewer suggests justifying the reason, if any, for this choice.
- >The referee warmly suggests specifying how the authors carried out the plasma processing of cells (i.e. w/o liquid, distance, etc. …)
- > Results: What does it mean exactly the terms 0-120 power?
- >Table 1. The authors show a list of the four human STS cell lines utilized in this study. Human Cell Line STS Subtypes Tissue of Origin References HT-10801 Fibrosarcoma Connective, SW-9821 Synovial Sarcoma Joint; Synovium [41] RD1 Rhabdomyosarcoma Muscle [42] 94T778 1,2 Liposarcoma Adipose [43] but proliferation, cell cycle, and apoptosis assays were performed only for 94T778 ones. Could you please explain the motivation for this choice? Why for FDA- the approved phase 1 clinical trial the authors focus their attention on this cell line?
- >2.1. Cell Viability: In the opinion of the reviewer it's not necessary to show a p-value if, because in the Materials and Methods Section it's claimed that differences were considered statistically significant for * p ≤ 0.05.
- >Figure 1: The reviewer suggests better specifying how the percentage was calculated. How do the authors specify the different behavior as a function of the He flow rate? Please include also the data of 94T778 similarly to other cell lines, including different cell density values.
- >Figure 3: in the opinion of the reviewer the nucleus shrinkage is not so evident thus the conclusions depicted are poorly supported.
- >Discussion, line 192: The modification of the histone mRNA has not been demonstrated in this study, please change the discussion accordingly
- >Materials and Methods, line 267: The reviewer suggests reporting the post-hoc test for ANOVA.
Round 2
Reviewer 1 Report
The manuscript was improved thanks to the revisions done. Thus the reviewer suggests publishing the paper in the present version without additional revision.
Reviewer 2 Report
Dear authors,
Thank you for answering to my questions. I would like just to stress to you that over the last 15 years, there have been hundred and hundred of in vitro studies which actually didn't really help to bring CAP to pre-clinical studies. In vitro studies that aim to prepare pre-clinical studies are an easy way to get a publication, but I personnally think it is time to move forward. And I still think that your study could have been implemented with preliminary in vivo data. I look forward to have these data published in a near future.